# Modeling of magnetic vestibular stimulation experienced during high-field clinical MRI
Ismael Arán-Tapia [1,2,3] ✉, Vicente Pérez-Muñuzuri[1,3], Alberto P. Muñuzuri [1,2], Andrés Soto-Varela [4,5,6], Jorge Otero-Millan[7], Dale C. Roberts[8] & Bryan K. Ward [9] ✉

## Abstract

**Background** High-field magnetic resonance imaging (MRI) is a powerful diagnostic tool but can induce unintended physiological effects, such as nystagmus and dizziness, potentially compromising the comfort and safety of individuals undergoing imaging. These effects likely result from the Lorentz force, which arises from the interaction between the MRI's static magnetic field and electrical currents in the inner ear. Yet, the Lorentz force hypothesis fails to explain observed eye movement patterns in healthy adults fully. This study explores these effects and tests whether the Lorentz force hypothesis adequately explains magnetic vestibular stimulation. **Methods** We developed a mathematical model integrating computational fluid dynamics, fluid-structure interaction solvers, and magnetohydrodynamic equations to simulate the biomechanical response of the cristae ampullares. Using high-resolution micro-CT data of the human membranous labyrinth, we ensured anatomical accuracy. Experimental validation involved measuring horizontal, vertical, and torsional slow-phase eye movements in healthy subjects exposed to varying magnetic field intensities and head positions. **Results** Our model accurately replicates observed nystagmus patterns, predicting slow-phase eye velocities that match experimental data. Results indicate that Lorentz force-induced stimulation of individual cupulae explains variability in eye movements across different magnetic field intensities and head orientations. **Conclusions** This study empirically supports the Lorentz force hypothesis as a valid explanation for magnetic vestibular stimulation, offering new insights into the effects of high-field MRI on the vestibular system. These findings provide a foundation for future research and improved clinical practices.

## Plain language summary

High-field magnetic resonance imaging (MRI) machines are used to provide detailed images of the human body to improve clinical diagnosis. However, they can sometimes cause dizziness and involuntary eye movements in people being imaged. These side effects are thought to occur because the magnetic field generated by the MRI interacts with the vestibular system, the part of the inner ear that helps maintain balance. To explore this further, we developed a computer model that simulates how these interactions occur. We then compared our simulation results with eye movement data from healthy individuals exposed to different MRI conditions. Our study confirms that the magnetic field generated by the MRI can trigger these symptoms. Our model allows these effects to be described more accurately, which could be used to improve safety and comfort during MRI scans in the future.

Magnetic resonance imaging (MRI) is a cornerstone of modern clinical diagnostics, with the strength of the magnetic field being a critical determinant of signal clarity and image resolution. However, the clinical utility of high-field MRI systems is accompanied by reports of dizziness and nystagmus among individuals near these devices, symptoms intensifying with field strength[1–3].

Located deep within the skull's temporal bone, the inner ear contains specialized sensors for rotation and gravity. These sensors detect head movements—angular movements via the semicircular canals and linear movements via the otoconial organs[4]. Hair cells within the inner ear are displaced by the movement of inner ear fluid relative to these accelerations, generating neural signals[5]. These signals are conveyed to the brain,

[1]Group of Non-Linear Physics, University of Santiago de Compostela, Santiago de Compostela, Spain. [2]Galician Center for Mathematical Research and Technology (CITMAga), Santiago de Compostela, Spain. [3]CRETUS Institute, Santiago de Compostela, Spain. [4]Division of Neurotology, Department of Otorhinolaryngology, Complexo Hospitalario Universitario, Santiago de Compostela, Spain. [5]Department of Surgery and Medical-Surgical Specialities, Universidade de Santiago de Compostela, Santiago de Compostela, Spain. [6]Health Research Institute of Santiago (IDIS), Santiago de Compostela, Spain. [7]School of Optometry, The University of California, Berkeley, CA, USA. [8]Department of Neurology, Johns Hopkins University School of Medicine, Baltimore, MD, USA. [9]Department of Otolaryngology-Head and Neck Surgery, Johns Hopkins University School of Medicine, Baltimore, MD, USA. ✉e-mail: ismaelaran.tapia@usc.es; bward15@jh.edu

providing it with information about the motion of the head. Additionally, these sensors are linked with reflexive eye movements that maintain visual stability during any head motion. The brain integrates the sensory input from both ears for balance. If there is an imbalance in this input, such as from stimulation of one ear, it may result in a repetitive eye motion known as nystagmus. Discrepancies among the various sensory systems that contribute to our sense of orientation can lead to sensations of dizziness.

The underlying mechanisms of dizziness in MRI machines are only partially understood. Prevailing theories implicate a Lorentz force, which acts on the vestibular apparatus, as a potential cause of these vestibular disturbances[6–8]. Yet, such theories do not account for all observed phenomena, particularly vertical nystagmus in healthy adults[6,9].

In this study, we develop a comprehensive mathematical model to simulate the biomechanical response of the cristae ampullares in the vestibular system when exposed to high-field MRI. Our model integrates computational fluid dynamics, fluid-structure interaction (FSI) solvers, and magnetohydrodynamic equations, using anatomically accurate data from high-resolution micro-computed tomography (μCT) scans. The model successfully replicates the eye movement patterns observed in healthy subjects, predicting slow-phase eye velocities (SPVs) consistent with experimental data across different magnetic field intensities and head positions. We show that the Lorentz force-induced stimulation of individual cupulae accounts for the variability in horizontal, vertical, and torsional eye movements. These findings support the Lorentz force hypothesis as a primary mechanism of magnetic vestibular stimulation (MVS) and provide new insights into how high-field MRI affects the vestibular system, with implications for safety guidelines and future MRI technology design.

## Methods
### Computational modeling and biomechanical simulations
**3D computational modeling of the membranous labyrinth.** We used a 3D reconstruction of a left membranous labyrinth, as shown in Fig. 1a. This reconstruction was obtained from a high-resolution μCT scan of a post-mortem human specimen available in an open-source database (http://www.earbank.org/ariadne.php), as described by David et al.[10]. We prepared this complex geometry for computational analysis using a procedure similar to the one detailed in our previous work[11].

In constructing our model of the membranous labyrinth, we treated the walls enclosing the endolymphatic fluid as completely rigid, except for the cupulae, which were modeled to reflect their elastic nature. The model includes the hair cell and dark cell regions that correspond with those observed in animals and humans[12,13] (Fig. 1a). We examined distinct head orientations, including four different ear-to-shoulder (ETS) positions to correlate our simulations with experimental observations of subjects within the MRI scanner, as depicted in Fig. 1b.

To simulate the right membranous labyrinth, we designed a symmetrical counterpart to the left and positioned it accordingly, as detailed in our recent work[11]. Both structures were discretized using consistent meshing protocols: a polyhedral mesh was applied to fluid regions, and a tetrahedral mesh to solid regions.

To facilitate data transfer between fluid and solid domains, we established mapped interfaces, effectively addressing the FSI problem between the endolymph (fluid) and the cupula (solid). However, for the electromagnetic domain, these interfaces presented limitations. The polyhedral and tetrahedral cell faces did not align, resulting in nonconformal boundaries that hindered the calculation of electric potential. To resolve this, we duplicated the geometry using a polyhedral mesh for fluid and solid regions without defining boundaries between them. This ensured mesh continuity and allowed for accurate calculation of electromagnetic properties throughout the entire membranous labyrinth.

Data related to Lorentz forces were mapped across the entire duplicated geometry and then transferred to the original polyhedral and tetrahedral cells defined for fluid and solid regions. Because the cell centroids or

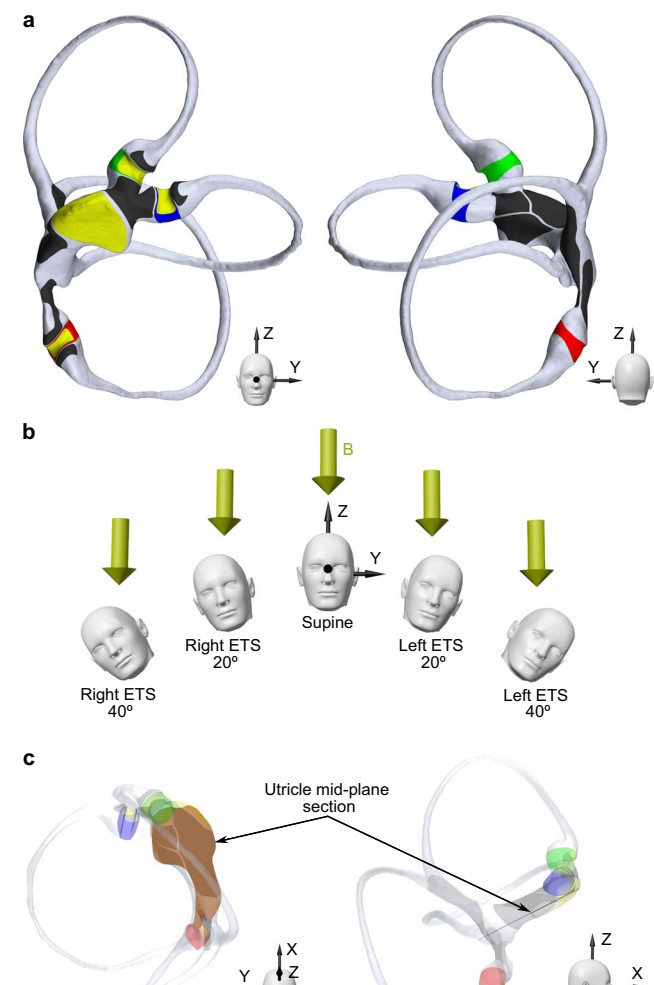

**Fig. 1 | Anatomical representation of the membranous labyrinth used for numerical simulations, different head orientations employed for statistical analysis, and the location of the utricle section used for measurements. a** Front and back views of the geometry and mesh model of the left membranous labyrinth when the head is in a supine position, featuring the rigid wall of the membranous labyrinth (gray) and elastic cupulae for the horizontal (blue), superior (green), and posterior (red) semicircular canals. Hair cell regions (yellow) can be distinguished in each crista ampullaris for each cupula and the utricular macula. Dark cell regions (black) are divided into each respective dark cell region for each ampulla, the utricular region, and the common crus region. **b** Head positions studied, showing the magnetic field direction in the MRI machine used in this study, oriented from the head to the feet. In the supine position, the Z axis (axial plane of the head) is aligned with the B field direction (Z axis of the bore); in this position, the utricle and the horizontal semicircular canal are ~20 degrees above the axial plane. For right and left ear-to-shoulder (ETS) positions, the membranous labyrinth is rotated 20 and 40 degrees relative to the B field direction. **c** The transverse section of the utricle's mid-region is presented relative to the membranous labyrinth. This plane was employed to determine the average electric current density, average Lorentz force, and maximum velocity for the model robustness analysis.

vertices of the original and duplicated geometry meshes did not coincide, the data were averaged to ensure a smooth transition. These data mapping techniques were carefully studied to ensure the accurate transfer of Lorentz forces throughout the simulation. Furthermore, we conducted a rigorous mesh independence analysis by creating progressively finer meshes and repeating the analysis until consistent results were achieved. This process ensured that we identified the optimal mesh configuration for accurate simulations of both fluid-solid dynamics and electromagnetic properties.

**Table 1 | Parameters for the endolymph and cupula used in the design of the electrodynamic and fluid-structure interaction models**

| Name | Variable | Value | Units | Reference |
|---|---|---|---|---|
| Electrical conductivity of the endolymph | $\sigma_e$ | 1.67 | S/m | Finley et al.[14] |
| Electrical conductivity of the cupula | $\sigma_c$ | 1.67 | S/m | Finley et al.[14] |
| Electric permeability of vacuum | $\mu_0$ | 1.26E-06 | H/m | Star CCM+ guide |
| Characteristic velocity of the endolymph | $U$ | 1.00E-04 | m/s | Measured |
| Characteristic length of the utricle | $L$ | 1.00E-03 | m | Measured |
| Endolymph density constant | $\rho$ | 1000 | kg/m$^3$ | David et al.[10] |
| Speed of sound in the endolymph | $c$ | 1500 | m/s | Star CCM+ guide |
| Compressible density of the endolymph | $\rho_e$ | $\rho + p/c^2$ | kg/m$^3$ | Star CCM+ guide |
| Dynamic viscosity of the endolymph | $\mu_e$ | 8.15E-04 | Pa.s | David et al.[10] |
| Density of the cupula | $\rho_c$ | 1000 | kg/m$^3$ | David et al.[10] |
| Dynamic viscosity of the cupula | $\mu_c$ | 8.15E-04 | Pa.s | David et al.[10] |
| Poisson ratio of the cupula | $\nu$ | 0.48 | | David et al.[10] |
| Young modulus of the cupula | $E$ | 4.26 | Pa | David et al.[10] |

**Quantitative modeling of endolymph and cupula dynamics.** In the study of MVS, the magnetic Reynolds number ($Re_m$) is a vital non-dimensional parameter that gauges the significance of a magnetic field in a moving conductive fluid. According to the parameters of the electrodynamic model used in this study (see Table 1), $Re_m$ is estimated to be on the order of $10^{-13}$, indicating a negligible value.

$$Re_m = \sigma_e \mu_0 UL \sim 10^{-13} \qquad (1)$$

In this equation, $\sigma_e$ represents the conductivity of the endolymph, assumed to be comparable to the conductivity of the cupulae $\sigma_c$[14]. The constant $\mu_0$ stands for the permeability of free space (also known as the vacuum electric permeability), $U$ denotes the characteristic velocity of the endolymph fluid movement and $L$ signifies the characteristic length of the utricle. The small $Re_m$ suggests that the magnetohydrodynamic forces generated by the fluid movement within the inner ear are minor compared to the magnetic fields typically used in MRI machines. Consequently, we focus on the Lorentz force as the principal magnetic force influencing the motion of the endolymph and the cupulae.

Hence, the electromagnetic model was only required at the initial time step of the numerical simulation to capture the static magnetohydrodynamic effect of the Lorentz force. This simplification reduces the mathematical complexity of MVS, eliminating the need for continuous mapping of information between electromagnetic and fluid-solid domains. Once the constant Lorentz force is determined, the model computes the associated FSI problem based on this force. This assumption allows us to simplify the mathematical representation of MVS.

Additionally, we operate within a quasi-static regime determined by the prescribed static magnetic field. This allows us to use a non-full-wave method, simplifying the electromagnetic problem by assuming that the fields are static or slowly varying based on Maxwell's equations. In this context, given the constant magnetic flux density $B$ in MRI and the absence of free charges within the endolymph and cupulae, the electric field $E$ is deduced by the gradient of the electric potential $\phi$:

$$E = -\nabla\phi \qquad (2)$$

Subsequently, the electric current density $J$ within the endolymph and cupulae are described by Ohm's law as the product of the electric field and the respective conductivity:

$$J = \sigma E \qquad (3)$$

Incorporating Eqs. (2) and (3), we derive the relationship for the current density in terms of the electric potential gradient:

$$J = -\sigma\nabla\phi \qquad (4)$$

The boundary conditions specified for the dark and hair cell regions (refer to Supplementary Data 1) allow the determination of the electric potential in the endolymph and cupulae domains. This potential can be utilized to solve Eq. 4 and ultimately obtain the electric current density. As an initial condition, the electric potential was set to 0.01 V[4], and this value has been maintained constant for the hair cell regions. Detailed calculations of these boundary conditions are provided in Supplementary Method 1, and an extensive model robustness analysis was performed to evaluate the effect of these chosen conditions (see Supplementary Method 2). The conclusions can be found in Supplementary Table S1, where certain variables were tested in the utricle mid-plane section shown in Fig. 1c. With the current density known, we can compute the Lorentz force $f_l$ across all endolymph and cupulae domains:

$$f_l = J \times B \qquad (5)$$

It is important to note that the Lorentz force values remain unchanged during the simulation because they depend solely on the boundary conditions. As a result, there is no need to persistently execute the Electrodynamic Potential Solver, which serves to minimize computational expenses.

The endolymphatic flow is determined by solving the Navier-Stokes equations. The resulting velocity $v$ of the endolymph is influenced by the Lorentz force as established in Eq. 5. The corresponding equation of motion for the endolymph is given by:

$$\rho e \frac{\partial v}{\partial t} + \rho e(v \cdot \nabla v) = -\nabla p + \mu_e \nabla^2 v + f_l \qquad (6)$$

Here, p represents the pressure within the endolymph, $\rho e$ denotes its compressible density and $\mu_e$ represent its dynamic viscosity. The equation factors in the Lorentz force to model the endolymphatic flow accurately.

In our model, the cupula regions are treated as elastic solids with linear and isotropic material properties that are nearly incompressible (Table 1). We allow for displacement only at the boundaries of the cupulae that interface with the fluid, while the remaining boundaries are fixed to adjacent structures. The displacement of the cupula $d$ is governed by the following dynamic equation:

$$\rho c \frac{\partial^2 d}{\partial t^2} = \nabla \cdot \sigma + f_l \qquad (7)$$

Where $\rho c$ is the density of the cupula, $\sigma$ is the stress tensor, and $f_l$ is the Lorentz force experienced by the respective cupula domain. This equation accounts for the forces exerted by the Lorentz force and the deformation of the walls in contact with the fluid, which are integral to our FSI model. Within the model, the pressures on the interfaces between the solid and fluid domains are reciprocally updated—fluid pressures influence solid deformation and vice versa—during each computational iteration to ensure accurate FSI simulation.

Following the calculation of the stress tensor $\sigma$, we can compute the shear strain component $\varepsilon_{XY}$ in the plane XY, which is critical for simulating the mechanical stimulation of the hair cells. This averaged shear strain $\varepsilon_{XY}$ measured at the crista ampullaris, rather than the cupula, serves as a more representative biomechanical indicator for the activation of the hair cells, in line with observations previously documented[11,15]. As a result, the crista can be excited or inhibited, which we have designated as positive and negative values, respectively. It is important to note that this does not imply a cessation of the information content of the afferent nerve signal. Excitation means that the firing rate of the afferent nerve fibers is above the resting value, while inhibition means that it is below the resting value.

We observed that shear strain reached a plateau at 30 seconds, as determined by a convergence criterion where subsequent stimulus increments did not exceed 1 μPa within a 2-second interval. At this steady-state level, the computed averaged shear strain XY in the cristae ampullares was used to derive the SPV. The SPV for each directional component—horizontal, vertical, and torsional—was calculated using the following relationships:

$$SPV_{hor} = LH^{ex} + RH^{in} - LH^{in} - RH^{ex}$$
$$SPV_{ver} = LS^{ex} + RS^{ex} + LP^{in} + RP^{in} - LS^{in} - RS^{in} - LP^{ex} - RP^{ex} \quad (8)$$
$$SPV_{tor} = LS^{ex} + RS^{in} + LP^{ex} + RP^{in} - LS^{in} - RS^{ex} - LP^{in} - RP^{ex}$$

Each variable corresponds to the magnitude of the absolute response from a specific crista, with 'L' denoting the left and 'R' the right membranous labyrinth, while 'H', 'P', and 'S' indicate horizontal, posterior, and superior cristae, respectively. The superscripts *ex* and *in* denote excitation and inhibition. The sum of these signals, informed by the known neurological pathways of the vestibulo-ocular reflex (VOR), yields the SPV[16]. Positive SPV values indicate a movement to the right, upward, and counterclockwise (top pole of the eye to the person's right side) whereas negative values suggest a movement to the left, downward, and clockwise for each corresponding eye movement component. Although inhibitory signals may be less influential than excitatory ones in the context of vestibular response[17], our model assumes a linear summation of stimuli, an assumption that holds for the lower frequencies characteristic of MVS. As a first approximation, our model also assumes that the VOR response is perfectly calibrated, that is, has a gain of 1, that the canals are perfectly perpendicular to each other, and that the shear at each crista has the same relationship to the head movements. This approximation will not hold exactly true for most subjects, but without having an additional extensive battery of tests, we decided to model this representative idealized scenario. Normalized SPV results at different head positions were adjusted to a normalized sinusoidal function to determine the adjusted R-squared, which provides an assessment of the goodness of fit of the regression model.

## Statistics and reproducibility
**Sample characteristics.** We studied five healthy volunteers (aged 24–41 years, four males) to capture three-dimensional binocular eye movements. These individuals were positioned supine in a darkened environment, and eye movements were recorded using infrared video goggles. We collected data in the Earth's magnetic field and within the bore of MRI scanners at different field strengths: 7 Tesla (Achieva, Philips, Hamburg, Germany), 3 Tesla (dStream Achieva, Philips), and 1.5 Tesla (Espree, Siemens, Munich, Germany). This research received ethical approval from the Johns Hopkins Institutional Review Board, and all participants

provided informed consent before their inclusion in the study. This study was conducted in accordance with the Declaration of Helsinki, and the protocol was approved by the Institutional Review Board of Johns Hopkins University School of Medicine (Approval No. NA_00041628). Informed consent was obtained from all individual participants included in the study.

**Infrared videooculography and head position measurement techniques.** Using MRI-compatible infrared videooculography goggles (RealEyes xDVR, Micromedical Technologies Inc.), we recorded subjects' eye movements at rest in a dark environment at a capture rate of 100 frames per second. The goggles, equipped with two Firefly MV cameras (PointGrey Research Inc., Richmond, BC, Canada), allowed for the dual capture of infrared imagery of each eye. We monitored real-time binocular eye position at a sampling rate of 100 Hz, assessing horizontal and vertical movements via pupillary tracking and torsional movements through iris pattern recognition. Ocular torsion angles were computed using a template-matching algorithm, which compared iris patterns against a baseline image taken with the subject's head upright at the start of the recording session. The methodologies employed for these measurements are elaborated in a prior publication[18].

We outfitted the goggles with an accelerometer (MPU-92/65 sensor, InvenSense) and a high-field magnetometer (MV2, MagVector, with a range of up to 10 Tesla) to concurrently track head tilt angles and magnetic field strength. These instruments were interfaced with an Arduino (Teensy 3.2) that fed data directly into the camera system. The accelerometer was calibrated to gauge the participant's head pitch relative to gravity, while the magnetometer measured the magnetic field intensity adjacent to the participant's right temple. The accelerometer continuously monitored the pitch angle, ensuring real-time consistency of the head pitch angle during scans. Due to the supine position of the participants rendering the accelerometer unable to detect lateral head tilts (ETS), we employed a non-metallic protractor for tilt measurements. The pitch and lateral tilts were synchronized with the centerline of the MRI table, extending from the chin through the nose's midpoint. We selected lateral tilt angles of 20 and 40 degrees, the latter being the maximum permissible within the spatial constraints of the 7 Tesla MRI bore.

**Integrated videooculography and head position calibration in MRI.** Participants were positioned supine on the MRI table, with heads aligned to a standardized pitch angle of approximately 100 degrees relative to Earth's horizontal plane. This alignment ensured that the lateral semicircular canals were elevated by about 30 degrees, consistent across trials. Head pitch angles were monitored throughout each trial. Before ETS tilt trials, a non-metallic protractor was employed for precise positioning. Visual stimuli were occluded using a double-layered black felt cloth, and eye movement baselines were established outside the MRI's magnetic field. Field strength near the participants' ears was 0.7 Tesla in the 7 T environment and below 0.5 Tesla in both 3 T and 1.5 T settings.

Trials commenced with subjects gazing directly ahead, with their gaze continuity confirmed by the experimenter. Data from eye movements were recorded continuously. After initial data collection outside the MRI bore, subjects were moved into the MRI bore for a 3-minute exposure to the MRI static magnetic field, followed by a 2-minute rest period outside the MRI bore to mitigate any residual effects. Each participant underwent three trials within a single session to ensure response stability. In the 7 T MRI, additional eye movement data were recorded at 20- and 40-degree lateral head tilt to both sides. Control experiments were conducted in an offsite lab, recording eye movements in the Earth's magnetic field with similar head tilts. Subjective vestibular sensations were documented post-experiment. Reports included transient rotational perceptions at 1.5 T in one subject, mild turning sensations at 3 T in two subjects, and rotational sensations at 7 T in all subjects, often described as the head rotating leftward and feet rightward. With ETS head tilts, participants described a perceived tilt in the rotation axis corresponding to the direction of the tilt.

**Quantification and normalization of nystagmus responses in MRI.** Using custom software, we identified the start and end of each nystagmus slow phase and applied an automatic least-squares fitting to the intervening data points. This process yielded a series of SPV data points. Ocular torsion was quantified according to the methodology detailed by Otero-Millan et al.[19]. Upon MRI entry, we observed an initial surge in nystagmus velocity, culminating in a peak before a partial central adaptation[20]. Focusing on the maximal response to avoid adaptive effects, we refined the raw eye movement data by removing artifacts from saccades, quick phases, and blinks. We then isolated the data to the temporal window encompassing the peak nystagmus and selected the ten highest SPV values for each eye and movement component. These values were averaged to ascertain the mean peak SPV for each nystagmus component per trial. From these ten values, the maximum and minimum were used to calculate the standard error of the mean peak SPV measurement. In addition, to isolate the MVS effect, we adjusted the SPV values by subtracting the mean baseline SPV recorded in Earth's magnetic field in a supine position. This correction ensured our findings reflected the MVS response, uncontaminated by any inherent nystagmus in darkness. We computed the average peak SPV for all subjects and trials within the same MRI scanner, along with the appropriate propagation of error, to establish a range of peak ocular velocities characteristic of the study population.

To align the experimental data with simulation outputs, we normalized the peak SPV values relative to the maximum value of each eye movement component. Vertical nystagmus was minimal when subjects lay in a head-neutral position; hence, the normalization of vertical SPV was based on the maximum horizontal SPV. The averaged values across all subjects were used to assess the fidelity of our experimental findings to the simulations at different B field intensities and head positions. To do this, we calculated the difference between the normalized SPV experimental and numerical data points, then determined the average of these differences as a percentage.

Regarding the orientation of the magnetic field vector B in MRI machines, it was oriented cephalocaudally in the 7 T and 3 T machines, with subjects entering the MRI in a supine, head-first position. In contrast, the 1.5 T MRI's magnetic field was directed caudocephalically, as expected in accordance with manufacturer's standard installation practice. This inversion was associated with nystagmus occurring in the reverse direction. To maintain consistency in comparing results across different MRI field strengths and considering simulations indicating that results should be mirror images, we inverted the sign of the SPV data collected from the 1.5 T MRI.

## Reporting summary

Further information on research design is available in the Nature Portfolio Reporting Summary linked to this article.

## Results

### Electrical currents and Lorentz force-induced dynamics in the inner ear

The streamlines of the electrical current depict the direction of the current flow throughout the entire membranous labyrinth, entering the endolymph from the dark cells that line the membranous labyrinth and flowing to regions of sensory hair cells (Fig. 1a). The hair cells of the cristae ampullares within the semicircular canals receive electric current from the dark cell regions located near their respective cristae. Additionally, dark cell regions in the common crus and the back of the utricle also contribute to the current, which flows through the entire semicircular canal (Fig. 2a). The currents flowing to the utricular macula primarily originate from the dark cell regions in the utricle. Those currents originating from the back of the utricle and common crus terminate in the macula's posterior part, while the others are distributed throughout the macula. As a result, the highest electric current density occurs in the center of the utricle, where most of the currents are directed to the front of the utricle.

When the head is exposed to a magnetic field, the complex distribution of currents interacts with the magnetic field (i.e., B field), generating a Lorentz force. The direction of the Lorentz force is expected to be orthogonal to both the B field direction and the electric current direction, with the latter varying depending on the specific region of the membranous labyrinth. For instance, the B field can be aligned along the Z axis from the head to the feet, and the electric current direction in the utricle is mostly in the X-axis from the posterior to anterior region of the utricle (Fig. 2a). This coupling generates a resultant Lorentz force that points approximately along the Y-axis from the right ear toward the left ear in both membranous labyrinths (see Fig. 2b), reaching its maximum at the center of the utricle where higher electric current density occurs.

The endolymph, initially at rest to simulate a head in a resting position, begins to flow due to the Lorentz force acting within it. This interaction is maintained while the magnetic field is present, like the conditions inside an MRI bore. The combination of Lorentz forces in the utricle generates higher fluid velocities in the middle, resulting in two vortices (Fig. 2c). The larger vortex is in the front of the utricle (maximum velocity 134 µm/s in a 7 T MRI). A smaller one with lower velocities (maximum 55 µm/s) is found in the back. In both labyrinths, the rotation direction is clockwise for the larger vortex and counterclockwise for the smaller. While the axis of rotation of the larger vortex is approximately along the Z axis, the axis of rotation of the posterior vortex is more longitudinally oriented along the utricle, extending into the common crus. Since there is electrical current throughout the membranous labyrinth, Lorentz forces appear in all areas but with lower intensity. For instance, fluid displacements occur within the semicircular canals and ampullae, with velocities three and two orders of magnitude smaller than in the utricle, respectively (the average maximum velocity for all canals is 0.3 µm/s, and for all ampullae, it is 2 µm/s).

The endolymphatic flow displaces the cupulae of all six semicircular canals. Although the fluid velocity quickly stabilizes within one second, a steady state of deformation occurs after approximately 30 seconds due to the viscoelastic properties of the cupula. Displacement of the cupula can be measured as shear strain XY in the crista ampullaris regions (Fig. 3). All the cristae ampullares exhibit the same shear strain pattern, with excitation/inhibition in the central zones and the opposite stimulus pattern in the periphery. We used the average shear strain pattern for each crista ampullaris to determine the overall excitation/inhibition for that crista ampullaris. The shear strain XY results of the cristae ampullares were opposite when comparing the right and left labyrinths (see Fig. 3). The net result is the deflection of all the cupulae of the right labyrinth toward the utricle (i.e., ampullofugal direction) and the left labyrinth away from the utricle (i.e., ampullopetal direction).

Reversing the direction of the magnetic field, such that it aligns along the Z axis from the feet to the head, inversely affects the Lorentz forces, now oriented in the negative Y direction within the central utricle area. Consequently, this reversal alters the rotational direction of the fluid vortices. Despite the change in direction, the magnitude of the resultant stimulus remains unchanged, although its vector sign is the opposite. This observation indicates that reversing the magnetic field direction will inhibit any crista ampullaris previously excited by the original field orientation and excite any crista ampullaris previously inhibited.

### Experimental analysis of eye movement responses to MVS

In this study, we quantified three-dimensional eye movements in an MRI to assess the effects of varying magnetic field strengths and head positions. Figure 4a presents a typical example of the normalized SPV values for nystagmus in one participant, indicative of a pattern consistent across all participants. The initial 20-second period represents the time required for the participant's entry into the bore of the 7 T MRI scanner in a supine position. The observed slow phase was predominantly leftward and counterclockwise (top pole of the eye to the person's right side), with negligible vertical movement, consistent with a null vertical SPV. Concurrent simulations estimated the average shear strain XY across the cristae ampullares, with the computational results corroborating the directionality of eye

**Fig. 2 | Illustration of the key physical parameters in magnetic vestibular stimulation. a** This panel shows electric current streamlines within the membranous labyrinth, projecting from the dark cells to the sensory hair cells, viewed from anterior and posterior perspectives. The streamlines are color-coded according to the associated vestibular end organ containing the targeted hair cells, with distinct colors for each crista ampullaris and the utricular macula. Representation of both scalar and vector components of (**b**) the Lorentz forces and (**c**) the endolymphatic fluid velocities within the transverse section of the utricle depicted in Fig. 1c. The flow pattern is characterized by two vortices generated by the Lorentz force effect: a stronger, clockwise vortex at the anterior part of the utricle and a smaller, counterclockwise vortex posteriorly.

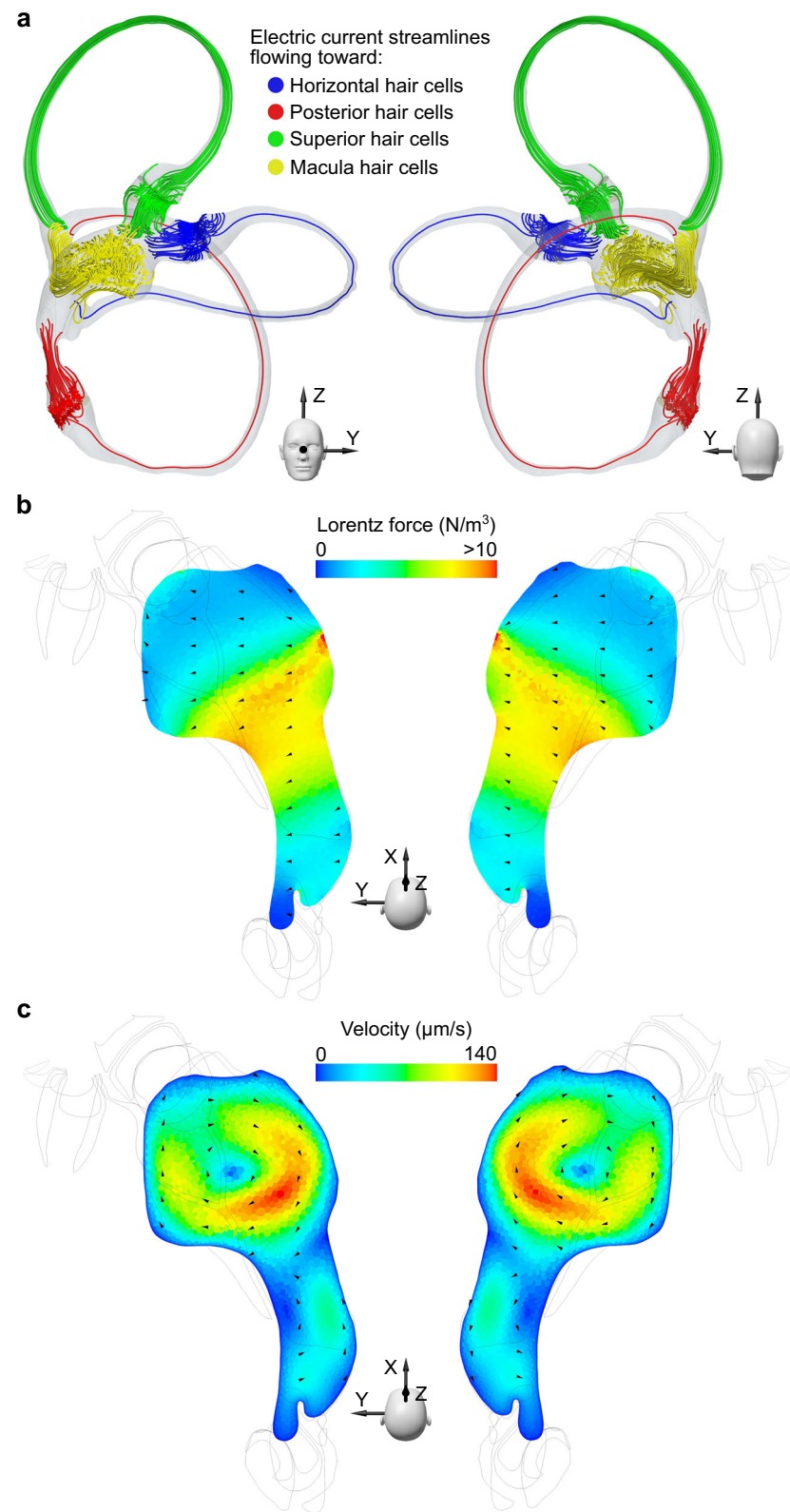

movements observed experimentally (as shown in Fig. 4a). Following a peak at approximately 30 seconds, the simulation values plateaued, whereas the experimental SPV measurements gradually decreased, reflecting neurological adaptation.

Analysis across three different magnetic field (B field) strengths revealed a proportional response in the horizontal and torsional eye movement components, with all subjects maintaining leftward and counterclockwise SPVs regardless of the strength of the magnetic field (top pole of the eye to the person's right side for the torsional component). In contrast, the vertical SPV did not exhibit an increase alongside higher B field strengths. Figure 4b summarizes the averaged SPV values for all participants.

**Fig. 3 | Mapping of shear strain XY across cristae ampullares.** Positive shear strain XY values indicate excitation (red), while negative values suggest inhibition (blue). Excitatory flow in the horizontal semicircular canals is characterized by movement toward the utricle. In contrast, excitatory flow for the superior and posterior canals occurs in the opposite direction. Colored arrows represent the overall direction of cupula displacement in this regard. The state of the cupula, whether excited or inhibited, is determined by the average shear strain XY across the entire crista.

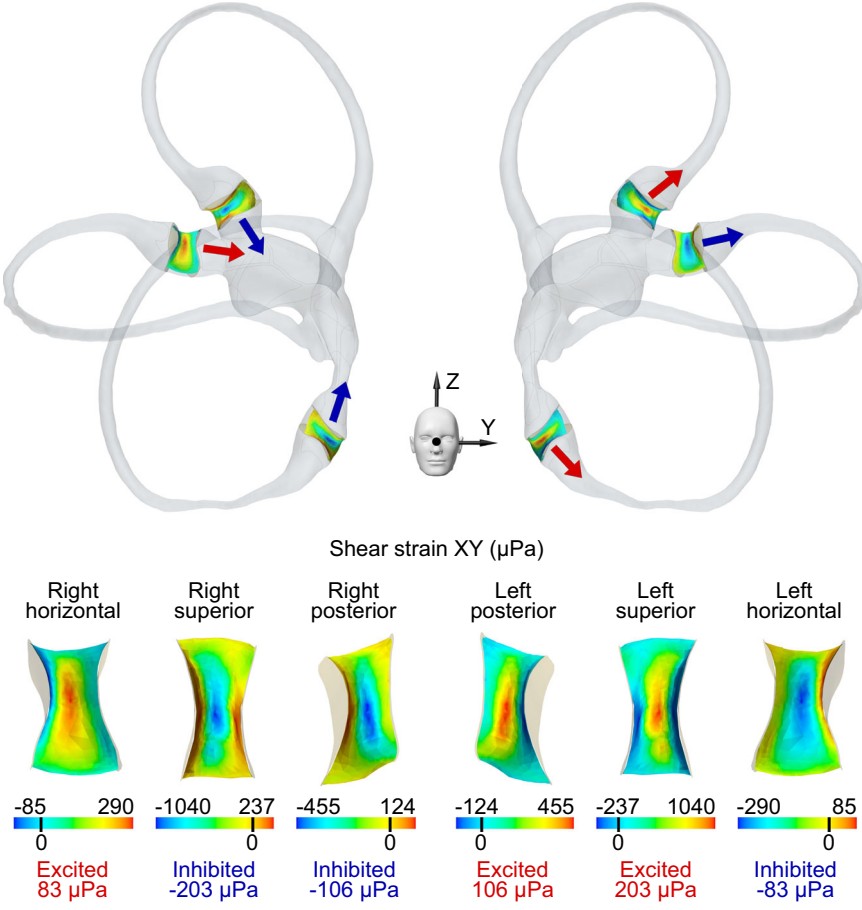

Different head orientations, specifically varying ETS positions, were analyzed to ascertain their effect on eye movement components. It was found that both the horizontal and torsional SPVs decrease as the head moves away from the supine position. In contrast, the vertical SPV exhibited directionally dependent behavior: upward SPV was recorded with the left ETS positions, and downward SPV with right ETS positions, with the transition through zero occurring at the supine position. These observations suggest that the vertical eye component behaves differently from the horizontal and torsional components.

The computational and experimental data demonstrated good agreement, as indicated by the average percentage deviation between the simulated and clinical SPV results shown in Table 2. For different B field strengths, the average percentage deviation is 1.8% for the horizontal component, 11.2% for the vertical component, and 3.2% for the torsional component. Similarly, for different ETS positions, the average percentage deviation is 25.8% for the horizontal eye component, 23.0% for the vertical component, and 12.2% for the torsional component.

## Simulated eye movement responses to theoretical head orientations

Through computational simulations, we evaluated eye movement responses by analyzing the averaged shear strain XY within the cristae ampullares for a range of theoretical head positions. Simulations aligned with the Z axis—coincident with magnetic field direction—revealed uniform responses across all cristae, mirroring the eye movement patterns observed for the supine position, as illustrated in Fig. 4c at a 0-degree angle.

Head rotation about the X-axis, simulating ETS tilts, was modeled to assess its effect on eye movements. The simulations predicted a sinusoidal pattern with a 180-degree wavelength ($R^2 = 1$) for all simulated eye movement components, providing a comprehensive representation of the normalized SPV over a 360-degree range of head positions along the X axis (Fig. 5a). The simulated horizontal and torsional components displayed antiphase behavior, neutralizing at -90 and 90 degrees, while the vertical component showed a 90-degree phase lag, resulting in zero values at 0 and 180 degrees. These null points for the nystagmus eye components are summarized in Table 3, obtained based on the shear strain XY values for each crista with their respective phase shifts, as detailed in Fig. 5c and Table 4.

Additionally, simulations of head rotations along the Y axis, representing flexion-extension (FE) movements, suggested that horizontal and torsional normalized SPV also fit a sinusoid with a 180-degree wavelength ($R^2 = 1$), but the vertical component remained null (Fig. 5b). The horizontal SPV reaches a null point at ~27 degrees of flexion and 153 degrees of extension, while the torsional SPV reaches a null point at ~50 degrees of flexion and 130 degrees of extension (see Table 3). These patterns were obtained from the shear strain XY measured in the cristae, which differed in phase and amplitude from those associated with ETS positions (Fig. 5d and Table 4).

## Discussion

Our findings demonstrate that electric current density within the membranous labyrinth is unaffected by the duration or strength of magnetic field exposure (Fig. 2a). The magnitude and distribution of these currents are governed by factors such as the electrical conductivity of the endolymph and the boundary conditions defined for the dark and hair cell regions. This would be influenced by sensory cell density for the dark and hair cell regions, as well as the relative electrical conductivity between the cupula and the endolymph (see Supplementary Method 2). Notably, a decline in hair cell numbers associated with aging[21] or other inner ear disease processes could diminish the overall electric current, thereby reducing susceptibility to MVS in older populations or those with peripheral vestibular disorders. Similarly,

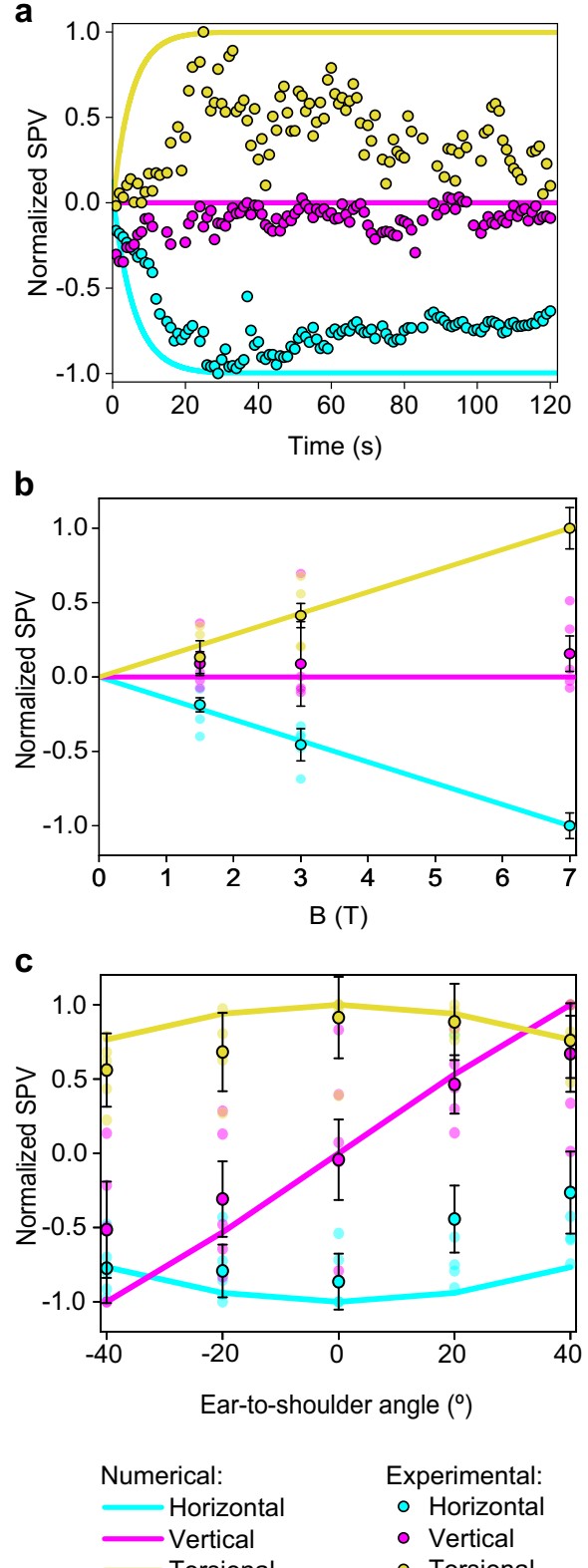

**Fig. 4 | Comparative normalized slow-phase velocity from numerical and experimental data.** Negative SPV values represent leftward, downward, and clockwise (top pole of the eye to the person's left ear) movements. In contrast, positive values represent rightward, upward, and counterclockwise movements for the horizontal, vertical, and torsional components of eye movement, respectively. **a** A comparison of normalized SPVs from computational analysis with experimental measurements from a representative participant inside the 7 T MRI scanner. **b** Aggregated normalized SPVs from computational and experimental data across varying magnetic field strengths while supine. **c** Normalized computational and experimental SPV peak data for different ETS head positions; negative angles denote right ETS, and positive angles denote left ETS, as elaborated in Fig. 1b. The vertical component SPVs in graphs **a** and **b** are normalized against the peak horizontal component values, while other components are normalized to their own peak values. Experimental values in graphs **b** and **c** reflect the average peak SPV across all participants after entering the bore (n=5), with error bars representing the respective standard error, reflecting inter-subject variability.

Numerical:
— Horizontal
— Vertical
— Torsional

Experimental:
○ Horizontal
○ Vertical
○ Torsional

variations in the ratio between the cupula and endolymph conductivity in the simulations correlate with alterations in electric current and, consequently, the vestibular stimulus experienced. This suggests that heightened sensitivity to MVS may be characteristic of individuals with a greater density of hair cells or increased electrical conductivity of the endolymph, aside from any neurological factors that may also be influential.

While more precise measurements of these parameters in humans could refine model accuracy further, we contend that the constant boundary conditions in our model effectively reflect the quantitative responses observed across different individuals without inner ear diseases (see Supplementary Method 1). This is because aging and natural variation in hair cell counts are expected to occur proportionally throughout the inner ear, and the electrical conductivity is only significant when there is a notable difference between the conductivities of the endolymph and cupulae. However, the literature typically considers these conductivities equal. It is important to note that using the same boundary conditions for different head orientations may not accurately capture the slight variations in current flow through the utricular macula caused by different deflections of the hair cells due to gravity[22,23].

This study's analysis reveals that an increase in Lorentz forces does not uniformly correspond to a heightened vestibular stimulus across different head positions. Instead, the anatomical nuances of the utricle, along with the specific origination points and terminations of electrical currents, play a pivotal role in determining fluid movement and the formation of vortices. The observed fluid dynamics within the utricle's vortices often defy the primary orientation of the Lorentz forces, showcasing movements in various directions, including counter to the force (see Fig. 2b, c). Previous models, such as the one by Antunes et al.[24], have not accounted for this intricate behavior, likely due to a less detailed representation of the inner ear's geometry than what we achieved with µCT[10]. Our approach, integrating a computational fluid dynamics (CFD) model with an FSI solver, captures the complex interactions leading to the stimulus experienced by all six cristae ampullares (Fig. 3). While our model also considers the magnetohydrodynamic electromotive forces arising from endolymph movement, their impact is minimal relative to the MRI's magnetic field, as indicated by the low magnetic Reynolds number outlined in our methods. Interestingly, the vortex locations identified in our stimulations correspond with artifacts recently observed in 3 T MRI scans, which have been hypothesized to stem from the movement of protons in these vortices[25].

Endolymph velocity within vortices quickly reaches equilibrium in less than a second, as shown in our findings. The cupula, however, exhibits a delayed response, achieving a stable deformation after around 30 seconds due to its intrinsic elasticity and restoring force[26]. This value is consistent with the predictions for a constant acceleration based on the dynamics of the endolymph in the semicircular canals[27]. This response, where the cupula retains this displaced state until the magnetic field is discontinued, causes constant nystagmus and simulates a constant acceleration[20]. Although such a response is detectable experimentally, neural adaptation causes a slow reduction in the nystagmus over time (Fig. 4a). It is also crucial to recognize that our simulation presupposes a consistently applied magnetic field, disregarding the transient effects experienced by subjects as they transition into the MRI's magnetic field. This entry process involves movement on the bed into the bore and through a gradient magnetic field, typically lasting ~20 seconds, which likely moderates the slope of the stimulus observed experimentally compared to our computational model (Fig. 4a).

**Table 2 | The average percentage deviation between numerical and experimental normalized SPV at different magnetic field intensities and ETS positions**

|  | Average deviation for B field variation (%) | Average deviation for ETS head variation (%) |
|---|---|---|
| $\Delta SPV_{hor}$ (%) | 1.8 | 25.8 |
| $\Delta SPV_{ver}$ (%) | 11.2 | 23.0 |
| $\Delta SPV_{tor}$ (%) | 3.2 | 12.2 |

The experimental data points used are those depicted in Fig. 4b, c, obtained from the average values of five subjects ($n = 5$).

**Table 3 | Null points for slow-phase components of nystagmus for Ear-to-shoulder and Flexion-extension head positions**

| Eye component | Ear-to-shoulder (°) | | Flexion-extension (°) | |
|---|---|---|---|---|
|  | 1° null | 2° null | 1° null | 2° null |
| Horizontal | −90 | 90 | −27 | 153 |
| Vertical | 0 | 180 | – | – |
| Torsional | −90 | 90 | −50 | 130 |

The head orientation measured in degrees corresponds to null points along the sinusoidal pattern for each component of nystagmus shown in Fig. 5a, b.

**Table 4 | Null points for the cristae ampullares stimulus for Ear-to-shoulder and Flexion-extension head positions**

| Ear-to-shoulder angle (°) | | | | |
|---|---|---|---|---|
| Crista ampullaris | Left ear | | Right ear | |
|  | 1° null | 2° null | 1° null | 2° null |
| Horizontal | −12 | 168 | 12 | 192 |
| Posterior | −19 | 161 | 19 | 199 |
| Superior | −42 | 138 | 42 | 222 |
| **Flexion-extension angle (°)** | | | | |
| Crista ampullaris | Left ear | | Right ear | |
|  | 1° null | 2° null | 1° null | 2° null |
| Horizontal | −27 | 153 | −27 | 153 |
| Posterior | −71 | 109 | −71 | 109 |
| Superior | −42 | 138 | −42 | 138 |

The head orientation measured in degrees corresponds to null points along the sinusoidal pattern for each cristae's shear strain XY responses shown in Fig. 5c, d.

In our study, we observed a linear response of the horizontal and torsional eye movement components to the strength of the magnetic field across all subjects, a pattern not mirrored by the vertical component, which typically approaches a null value (Fig. 4b). This phenomenon is elucidated by the computational simulation outcomes for each crista (Fig. 3). Specifically, the left horizontal cupula is inhibited. At the same time, the right is excited, which amplifies the leftward horizontal eye movement in proportion to the magnetic field strength. In contrast, the vertical eye movement component is neutralized due to opposing responses from the right superior and posterior cristae and the left superior and posterior cristae. The net torsional eye movement component results in a counterclockwise movement. The differential response is detailed in our mathematical model (see Methods). Notably, the strong concordance between our simulations and the experimental data shown in Table 2—with an average deviation of just 5.4% for all eye components—validates our model's capacity to predict the vestibular response to varying B field intensities. These findings are consistent with the predictions of Lorentz force theory and corroborate the patterns observed in previous clinical research and the current study[6,24]. Although we focused on the peak values of nystagmus obtained inside the scanner, all data collected before, during, and after exposure to the magnetic field were consistent with our previous studies. The data showed a reversal of the nystagmus upon exiting the magnetic field[28], and the peak response was delayed due to the time taken for entry and exit from the MRI bore[29].

Outside the MRI bore, the magnetic field (B field) strength is typically less than 1 T for 1.5 T and 3 T MRI machines but can approximate 1 T near 7 T machines[30]. It might be presumed that such exposure could lead to an adaptive response, potentially obscuring a more pronounced stimulus upon entering the 7 T field. However, our experimental data did not show a significant adaptation effect that would alter the peak nystagmus measurements inside the bore. A more influential factor in our experimental observations was the baseline nystagmus data collected while lying in darkness away from the MRI. Subtracting this baseline nystagmus was particularly relevant when the stimulus was small, as it markedly affected the determination of the vertical component's null point across various B field strengths. This step also allowed us to compare the peak response upon entering the bore using normalized values referenced to the same baseline.

In our computational model, the left and right membranous labyrinths are symmetrical along the sagittal plane, leading to mirrored responses in the left/right cristae (Fig. 3) and when simulating head-first versus feet-first

entry (i.e., reversing the B field direction). Anatomical variation between the actual left and right labyrinths in individuals[31,32] could account for the lack of symmetry in some participant responses. Such inter-individual variability is likely responsible for the range of responses observed across subjects, as reflected in the error bars in Fig. 4, which reflects the inter-subject variability peak nystagmus recorded after entering the bore ($n = 5$). See the Methods section called Quantification and normalization of nystagmus responses in MRI.

Unlike in the supine position, asymmetries arise in ETS orientations due to different alignments of the left and right membranous labyrinths relative to the magnetic field. This causes the Lorentz forces to change their direction in the equivalent points of one labyrinth relative to the other. These asymmetries likely caused the observed decreases in the horizontal and torsional eye movement components when the head was positioned away from supine. In contrast, the vertical eye component increased, with upward movements in left ETS and downward in right ETS positions, as shown in Fig. 4c. The computational model results matched these experimental observations (see Table 2), with an average deviation of 20.4% for all eye movement components, supporting its predictive capability for SPV eye movements across various head orientations. Detailed analysis of the average stimulus of each crista ampullaris, as presented in Fig. 5c, clarifies these effects. The horizontal and posterior cristae responses were more excitatory in right ETS positions, while the superior cristae were more inhibitory, with inverse effects noted in left ETS orientations. This pattern accounts for the vertical eye movement component behaving inversely to the torsional one since the superior and posterior cristae contribute differently to the vertical response but similarly to the torsional response within the neurological pathways.

Our numerical simulations reveal that the stimulus to each cupula is characterized by a sinusoidal pattern with a 180-degree wavelength. This consistent pattern across all cupulae has allowed us to create a predictive model for SPV eye movement responses to ETS and FE head positions, as depicted in Fig. 5. The neurological integration of these stimuli, involving the simple addition or subtraction of the outputs from each crista, results in SPV eye movements that consistently exhibit a 180-degree sinusoidal wavelength. These findings are consistent with experimental observations for various ETS orientations reported by Mian et al.[9]. However, our study diverges from the suggestion that FE movements follow a sinusoid of a different wavelength. Corroborating this experimentally requires data not currently obtainable, given the physical geometrical constraints of the human body and MRI machines.

The simulations provide a method to pinpoint the null points of eye movement components by summing the sinusoidal outputs from each

**Fig. 5 | Computationally derived normalized slow-phase velocity of eye movements across various head orientations and the respective cristae ampullares stimulus. a** SPV profiles for head tilts simulating ETS movements, where negative values mean head-oriented right ETS, and positive values mean left ETS. **b** SPV profiles for head movements in FE, where negative values mean head flexion and positive values mean head extension. **c** Cristae ampullares stimulus profiles for ETS positions. **d** Cristae ampullares stimulus profiles for FE positions. Positions for the null stimulus for each directional component of nystagmus and each cristae's shear strain can be found in Tables 3 and 4.

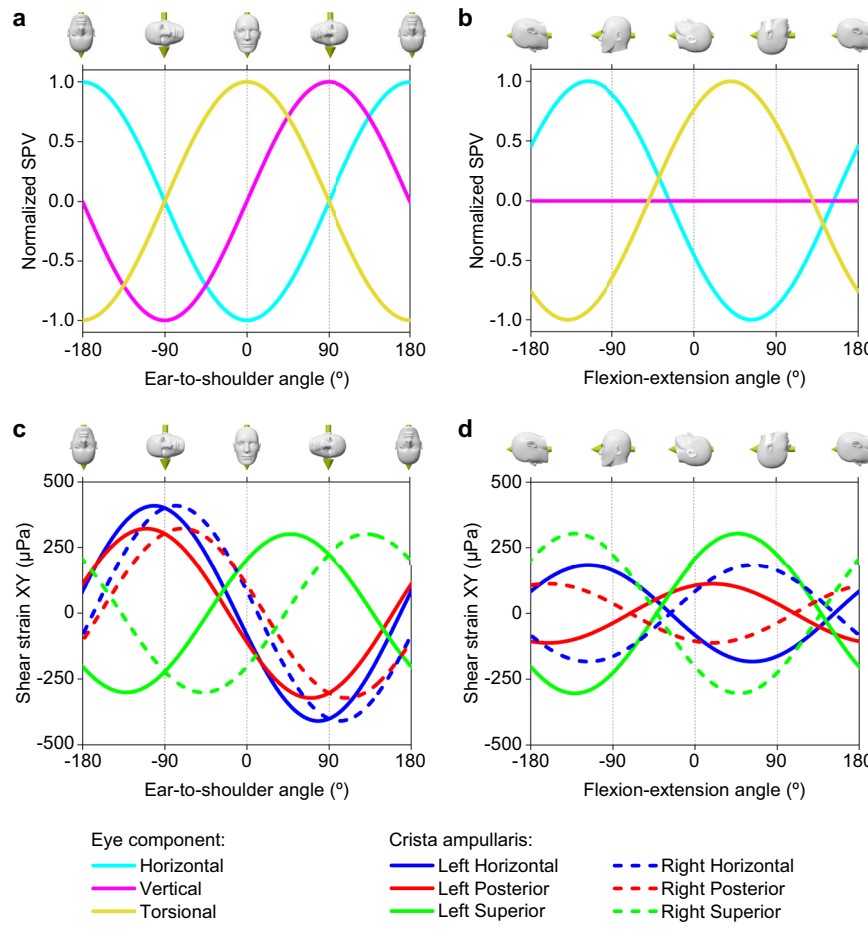

crista. This approach elucidates the observed phenomena wherein null points of eye movements during ETS and FE positions occur at different head angles. For ETS movements, the responses of each pair of cristae follow a sinusoid de-phased an equal distance with respect to 0 degrees but diverge in direction (see Fig. 5c and Table 4), resulting in null points for eye movement components at 90-degree intervals, as shown in Fig. 4 A. Conversely, in FE movements, the sinusoids of corresponding pairs of cristae in both ears are in opposed phases, leading to specific null points for each crista pair at the same head position (see Fig. 5d and Table 4). Notably, our model predicts that during FE, the horizontal SPV component reaches a null point at ~27 degrees of flexion, which may correlate with the alignment of the utricular macula relative to the magnetic field (Roberts et al. supplementary material[6]). However, the torsional SPV null point occurs at 50 degrees of flexion, suggesting that it is not possible to achieve a complete null response to the MVS at a specific head position.

Our results indicate that a null response in eye movements does not equate to an absence of stimulus in the corresponding cristae. Except for the horizontal SPV component in FE, where the null point truly corresponds with null cristae ampullares stimulus in both horizontal cupulae (Fig. 5b, d). For all the other SPV components studied, the null point is achieved for a state of perfect balance between the corresponding cupulae. For example, a null vertical SPV observed in FE head positions results from a precise counterbalance of stimuli from the four posterior and superior canal cristae in the VOR pathway[16], not from a lack of stimulus in these cristae. Therefore, while the Lorentz force continually influences the endolymph within the membranous labyrinth, at specific head orientations, the resulting cupula displacements due to endolymph movement reach a state of perfect balance.

It is important to acknowledge that the numerical simulations are based on anatomic data from a single individual, which may accurately reflect the specific vestibular responses of that individual in an MRI machine but may not be generalizable to all individuals. The necessity for post-mortem micro-CT imaging to achieve the requisite image quality for precise MVS modeling currently precludes in vivo validation. Despite this constraint, the use of normalized values in our study offers a generalizable understanding of the stimuli affecting the cristae ampullares across a broad population segment, as evaluated in our model robustness analysis. Furthermore, these normalized metrics facilitate direct comparisons between the shear strain-induced SPV predictions and actual SPV measurements from live subjects undergoing MRI. The congruence of our findings with the directional variability of the magnetic fields lends robust support to the Lorentz force hypothesis, increasing proportionally to the B-field intensity and responding to a sinusoidal function when evaluated at different head orientations relative to the B-field direction. It correlates with clinical phenomena such as the inversion of nystagmus when the subject's orientation is altered within the MRI[6]. This numerical model is an important step forward, offering novel physical and mathematical perspectives on MVS, and it holds the promise of shedding new light on the complexities of vestibular function and dysfunction.

## Data availability

All main data generated or analyzed during this study are included in the figures and tables of this published article, along with supplementary files. The supplementary information file provides additional details on the boundary conditions (Supplementary Method 1) and the variables selected for the model robustness analysis (Supplementary Method 2). The parameters defining the boundary conditions of the electrodynamic potential model are available in Supplementary Data 1, and the source data for Fig. 4 are available in Supplementary Data 2. Additional data are available from the corresponding author upon reasonable request.

## Abbreviations

| | |
|---|---|
| Magnetic vestibular stimulation | (MVS) |
| magnetic resonance imaging | (MRI) |
| computational fluid dynamics | (CFD) |
| fluid-structure interaction | (FSI) |
| ear-to-shoulder | (ETS) |
| flexion-extension | (FE) |
| micro-computed tomography | ($\mu$CT) |
| slow-phase velocity | (SPV) |
| vestibulo-ocular reflex | (VOR). |

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

## Acknowledgements

We want to thank Hanzhang Liu, Adrian Paez, and Cindy Maranto for their help in coordinating MRI scanner time and David Zee for his thoughtful reading of the manuscript. Images were created using GNU Image Manipulation Program (GIMP), Simcenter Star CCM+, Blender, and MATLAB. Editing for clarity was assisted by Grammarly and ChatGPT 4.0. The simulations were run in the Supercomputer Center of Galicia (CESGA), and we appreciate their support. We also acknowledge the funding provided by various entities that supported this study. This research is supported by the Spanish Ministerio de Economía y Competitividad and the European Regional Development Fund, through research grant PID2022-138322OB-100, funded by MCIN/AEI/10.13039/501100011033 and by "ERDF A Way of Making Europe." It is also supported by Xunta de Galicia under research grant no. 2021-PG036, the Instituto de Salud Carlos III (ISCIII) under grant number PI23/00248, and the National Institutes of Health (NIH) K23DC018302.

## Author contributions

I.A.T. designed the models, performed the CFD simulations, analyzed the results, and wrote the original draft. B.K.W. participated in study design, data collection, and writing the original draft. J.O.M. assisted with data analysis. D.R. contributed to data collection. All other authors (V.P.M., A.P.M., and A.S.V.) participated equally in supervision and conceptualization. All authors approved the final version of the manuscript.

## Competing interests

The authors declare no competing interests.
