## [Transparent Peer Review file · Communications Medicine]

Modeling of magnetic vestibular stimulation experienced during high-field clinical MRI

Corresponding Author: Dr Ismael Arán-Tapia

Version 0:

Reviewer comments:

Reviewer #1

(Remarks to the Author)

The authors present a fluid dynamic model of the labyrinth, supported by micro-CT data from a post-mortem labyrinth to numerically simulate the magnetic vestibular stimulation (MVS) effect and resulting three-dimensional eye movements. The model is then compared to eye-movement data from 5 participants measured in three different magnetic field strengths, and two roll, or ear-to-shoulder rotations for each side. Several interesting findings result from this model. The electrical current across the labyrinth do not change over time in a static magnetic field. These currents are rather affected by factors that were not systematically manipulated in this study. The endolymphatic fluid velocities across the utricle show two vortices and not one. And the resulting eye movements vary sinusoidally with the head position, resulting from the push and pull of opposing currents in the three ampullae and assuming idealized structures. The work is an important step in understanding how the inner ear structures behave in a strong magnetic field. However, the manuscript was challenging to read in its current form and several questions remained.

The organization of the manuscript made it difficult to read. The methods were at the end, but the results did not include enough of the methods to be able to accurately interpret them until the results sections were read. Indeed, the main output of the study is the model, which requires one to go through the supplements as well before understanding what was done. The discussion is also quite long, and I felt I finally understood the goal of the manuscript after reading the discussion. I would suggest reworking the text such that a reader with little information about the project is lead through the work.

Also, I have a few concerns with the work itself that should be addressed before publication. These include:

My main concern is that the entire work is based on one model, with one set of boundary conditions and anatomical measures from a single labyrinth. The model is not fit to any data but uses normative values (when possible) from the literature, and anatomical and morphological information from the CT data. The resulting slow phase velocities were also not statistically compared to the experimental data. Here are my issues in detail.

The authors claim that their micro-CT data has a better resolution than currently published data (David et al. 2016 – Ariadne software). But both this study and theirs claim sub 20 micrometer resolution, so why is their scan so much better than previous ones. Also – there is an open-source database with 23 human labyrinths measured with clinical-CT and micro-CT, which would provide more variability and allow for better normative anatomical and morphological values (Wimmer et al. 2019 -<https://doi.org/10.1016/j.dib.2019.104782>) – why was this openly available data not used? What was the rationale for a using a single labyrinth instead?

Why was the known variability in anatomy of the labyrinth not used to demonstrate robustness of the model, or at least to show that for the range of known anatomical variability, the model provides meaningful results. This would strengthen the model and show that it may have more clinical utility beyond the MVS effect.

This previously published model (David et al. 2016) provides the ability to calculate shear strain in the ampullae which were used for calculating the resulting eye movements, why was this model not used, or at least compared to the model the authors propose, so that authors could demonstrate the superior nature of their model?

Why were the boundary conditions not manipulated, also to demonstrate robustness of the model? The manuscript in its current form is of limited utility because it is entirely dependent on the single model that the authors present. The model has many parameters, and it is unclear how the behavior would change if the parameters were varied.

Finally, I am unsure of the actual clinical value of the work. The work was done to explain the MVS effect. This effect is not

particularly clinically relevant. Individuals may complain about dizziness at the beginning of an MRI, but this typically disappears quickly. What other clinical relevance does this work have?

Minor comments:

Figure 1 B and C requires a great deal of imagination or spatial reasoning to envision the orientation of the utricle in relation to the original labyrinth figure. It would be helpful to have a bit more structure of the labyrinth in the background to understand the location of the forces and velocities better.

How old was the labyrinth? In the supplements I read 55 years but not specifically for the age of the labyrinth. Please make this clear. Are any structural changes expected at that age or from the preservation technique that would affect the results?

Consider adding LTS and RTS to the list of abbreviations, instead of simply putting them in the text.

The authors say that they did a 3 minute scan of the participants. What data were collected and was it relevant for the study? Were the eye-movements measured during this time, or after, and if they were measured during, would the resulting magnetic fluctuations affect the SPV?

R2 was used in the results, I am assuming to compare the simulations to a sinusoidal curve (since R2 was 1). This was not mentioned anywhere in the manuscript, please make this clear.

In the interest of open science, it would be helpful to make your model open source, or combine it with existing resources like Ariadne, please consider.

Reviewer #2

(Remarks to the Author)

General Comments

The authors describe a fully detailed simulation of the electrical and mechanical properties of the vestibular function in response to an externally applied magnetic field. The work takes the level of detail another quantum leap in terms of accuracy and conclusions. I would liken this to a jigsaw, where so far we have the frame (observations) and just some of the pieces in place. Enough to know that the Lorentz force is the probable driver of the experiences of subjects in and around MRI. This work really does fill in the remaining pieces to join up the bits we have with a fully reliable predictive model without significant assumptions. So the electromagnetic model is performed at high spatial accuracy using latest available resolution micro-CT, recent reliable values of the stria currents and boundary conditions have been utilised, a suitable flow/dynamic and elastic model has been used – all for the first time. The latter demonstrates the presence and relevance of the vortex flow, hitherto unproven. This work is a tour-de-force, scientifically sound, well written and presented.

Does the manuscript have technical or conceptual flaws that should prohibit its publication? No

Are the conclusions original? Yes

Do you feel that the results presented are of immediate relevance for many people in your own field or for a broader audience? Yes, both. And all those interested in vestibular function, balance, physiology, fMRI, biophysics, electromagnetic field safety.

If applicable, does the manuscript reporting follow ICMJE and EQUATOR recommendations? Yes

If you recommend publication, please outline briefly what you consider to be the outstanding features: See above general comments

Specific Comments on Manuscript

P5 lines 94 to 105 section. Some repetition and needs a bit of re-ordering and re-writing. You also seem to imply that the magnetic field is always head-feet – specifically supine subject in conventional MRI. This is the example but the Lorentz force ideas need explaining more generally. Also say head at rest in the section – so assumed steady state.

P5 L106 I would prefer the magnetic field to be 'present' rather than 'active'. To me 'active' seems to indicate that we can switch it on and off at will like an evoked response stimulus.

P6 L125 (and elsewhere). Probably technically OK to use inhibit/excite but in actuality there is a signal both ways of deflection. I think that these words, to a general reader might confuse – thinking that the information content of the afferent nerve signal stops in some way. Can a very brief reminder be added.

P21 L386 '... given the physical geometrical constraints of the human body and MRI machines.' Or something similar.

P28 L563 Add somewhere that this electrodynamic problem is firmly in the quasi-static regime and can be solved with non-full-wave EM finite-element or boundary-element methods. Also, you didn't actually say how you did this simulation from a computational point of view.

P31 L637 The canals and cupulae are not-orthogonal, so your statement here can't be true. But presumably you are arguing that within natural variation of subject response, this can be assumed.

P33 L691 'scan' is not what you are doing (I hope with all that equipment in there!). Use 'exposure' or 'trial' as elsewhere.

P35 L738 add '...as expected in accordance with manufacturer's standard installation practice'. It isn't a random thing or mistake.

P34. You quite rightly discuss baseline correction of data. Any comment on the fact that the subject has already adapted to the magnetic field present (0.7 T for 7T magnet). The additional field exposure would be measured at peak response, but the 0.7T is measured at the lower steady state. Would be about 5% error perhaps. Just that you are subtracting out 50 microtesla effects – which are presumably just resting-state subject effects.
And that is all I can find!

My recommendation is that the article should be published subject to minor review of points made above.

Version 1:

Reviewer comments:

Reviewer #1

(Remarks to the Author)

Dear Authors,

Thank you for your revision of the manuscript. I believe that it has significantly improved. The details of the model are helpful to readers to appreciate the relevance of your work. I believe the relationship between head position and SPV for the three eye movement directions, the fluid dynamic model itself and what the model says about factors that may influence SPV are novel contributions that will be important for basic and clinical work. Also the figure with the orientation of the utricle is now much easier to understand. I believe the work warrants publication in Nature Communications Medicine. Thank you for your work.

Reviewer #2

(Remarks to the Author)

My overall opinion of the paper after revision has not changed. It is a significant and insightful work and should be published.

The authors have replied, corrected or altered the manuscript to my satisfaction and I am not going to request any further changes.

I can see and fully understand the reasoning of the first referee, and hence the questions and points raised are all valid. Again, I believe the authors have fully responded with changes, or made arguments for non-changes where appropriate. I fully agree with the authors' analyses in such cases, particularly the difficulty of getting variation data on the labyrinth structures and materials. Such parameters can only be approximate within calculated bounds. The authors are using the best and most up-to-date data available in order to construct a model. The work should be published without further changes.

The Biophysics of Magnetic Vestibular Stimulation: Clinical Insights from High-Field MRI

Reviewer #1

Major comment

We appreciate your interest in our research findings and your valuable feedback, which we believe has helped to facilitate the understanding of our research and provide more evidence of our results.

Considering the general comments provided, we have implemented significant improvements throughout the document per your suggestions. Specifically, we have repositioned the methods section before the results section and integrated the extended data figures and tables into the main text, aligning with the correct structure for the journal. We anticipate this new arrangement will enhance readers' understanding of our study, making the results sections more accessible. Additionally, we have introduced a new figure (Figure 1C) and tables (Tables 2 and 3). Moreover, we have revised the discussion section to convey our observations whenever feasible succinctly.

Regarding the manipulation of factors affecting electric currents and the sensitivity of our results to a single model anatomy, we have conducted a robustness study of the boundary conditions and explained how these limitations could affect our results. Please find more detailed answers below. We hope these modifications address substantial concerns and meet the publication requirements.

Responses to the reviewer's comments

Comment 1: My main concern is that the entire work is based on one model, with one set of boundary conditions and anatomical measures from a single labyrinth. The model is not fit to any data but uses normative values (when possible) from the literature, and anatomical and morphological information from the CT data. The resulting slow phase velocities were also not statistically compared to the experimental data. Here are my issues in detail.

Answer 1: We understand your concern about our study's limitations. We have been attempting to address the limitation of using a single CT model over the years while designing mathematical models of the vestibular system. Unfortunately, we have been unable to obtain other micro-CT models of the human membranous labyrinth that could be useful for simulation purposes. Please refer to Answer 2 for more details regarding this issue.

We acknowledge that the manuscript did not adequately examine a set of variables used as inputs for the model. Please see our response in Answer 5 for more information regarding the model robustness analysis we performed.

Finally, our model determines the nystagmus slow-phase velocities (SPV) generated from shear strain XY on each crista ampullaris, resulting in values

measured in pressure units (Pascals). Meanwhile, the experimental eye SPV is measured in degrees per second. A direct statistical comparison between these two data sets is not feasible due to the unit difference. This was why we normalized both SPV values, allowing us to understand how much the model predictions deviate from the experimental data. The results of this statistical analysis were described for different magnetic field intensities and head-to-shoulder positions; refer to Paragraph 4 of the results section, "Experimental analysis of eye movement responses to magnetic vestibular stimulation." Additionally, we decided to reorganize these statistical results in a new table to enhance its visibility, see new Table 1. We clarified the procedure followed to determine the deviation between experimental and theoretical results in Paragraph 2 of the methodology section titled "Quantification and normalization of nystagmus responses in MRI."

Comment 2: The authors claim that their micro-CT data has a better resolution than currently published data (David et al. 2016 – Ariadne software). But both this study and theirs claim sub 20 micrometer resolution, so why is their scan so much better than previous ones. Also – there is an open-source database with 23 human labyrinths measured with clinical-CT and micro-CT, which would provide more variability and allow for better normative anatomical and morphological values (Wimmer et al. 2019 -<https://doi.org/10.1016/j.dib.2019.104782>) – why was this openly available data not used? What was the rationale for a using a single labyrinth instead?

Answer 2: Thank you for pointing out this limitation. Firstly, it is important to note that the micro-CT we employed is the same as the one used by David et al. (2016) – Ariadne software. We aimed to emphasize the accuracy of the geometry we

utilized by referencing the voxel resolution provided in their manuscript. However, considering this detail might cause confusion, we omitted this brief explanation. Please refer to Paragraph 1 of the "3D computational modeling of the membranous labyrinth" methodology section for further information.

While we were aware of the existence of the Wimmer et al. 2019 study, their open-source dataset only contains files for bony labyrinths, which are insufficient for accurately determining the fluid and solid dynamics necessary to solve the magnetic vestibular stimulation (MVS) problem. Unfortunately, we did not find other open-source datasets that included human membranous labyrinths when we performed this study. Currently, we are searching for new human membranous labyrinth models, ideally with similar micro-CT quality as presented by David et al. 2016. We hope to simulate them in the future and understand the effect of anatomical variability on the crista ampullaris stimulus, not only for MVS but also for other forms of vestibular stimulation.

In addition to this clear limitation, there are other computational and licensing constraints. These simulations took several weeks to run on a supercomputer, incurring associated costs and licensing requirements. The complexity of solving these Finite Element Method (FEM) and Finite Volume Method (FVM) models contributes to this time-consuming process. This complexity likely explains why other prior publications also utilized a single labyrinth model (Wu et al., 2012 - <https://doi.org/10.1016/j.heares.2021.108282>) or even simplified their models to include only a single semicircular canal connected to an ideal utricle (Goyens et al., 2019 - <https://link.springer.com/article/10.1007/s10237-019-01160-2>). Our model, which is not only based on micro-CT data resulting in a more complex

geometry but also addresses the electromagnetic problem and designing of electric boundary regions, represents a higher level of complexity.

The rationale for using only one labyrinth model can be explained by the lack of available models of the human membranous labyrinth and the resource intensive nature of the simulations. Despite the limitations, we showed the potential robustness of using a single membranous model to represent the biophysical behavior of MVS, especially when considering normalized SPV, as shown in Figure 5. For further details, please refer to Answer 5. We have described these limitations in the manuscript to explain why we used a single geometrical model in this study. Although we have clinically validated the numerical results from the nystagmus eye components, the model has not yet been tested in other anatomical geometries. Refer to the last paragraph of the discussion section for the updated text.

Comment 3: Why was the known variability in anatomy of the labyrinth not used to demonstrate robustness of the model, or at least to show that for the range of known anatomical variability, the model provides meaningful results. This would strengthen the model and show that it may have more clinical utility beyond the MVS effect.

Answer 3: Thank you for this interesting question. We do not believe that using known anatomical variability from the literature would be helpful for us. This is because we require an entire anatomical model of the membranous labyrinth to study how the nonlinear behavior of fluid dynamics affects vestibular stimulation. For instance, just the appearance of vortices in the utricle would make it challenging to obtain a direct correlation by comparing the variability in the

orientation of the semicircular canals with the MVS responses. For further details, please refer to Answer 4.

This was why we chose to complement this mathematical study with a clinical experiment to support our findings. This validation gave us confidence that the model accurately captures the qualitative behavior of MVS-induced nystagmus due to the Lorentz force by using normalized shear strain XY in the crista ampullaris. Furthermore, this allowed us to anticipate that other subjects would exhibit similar qualitative behavior to what we observed in our numerical and experimental results, increasing proportionally to the B-field intensity and responding to a sinusoidal function when evaluated at different head orientations relative to the B-field direction. We include the updates in the last paragraph of the discussion section.

Comment 4: This previously published model (David et al. 2016) provides the ability to calculate shear strain in the ampullae which were used for calculating the resulting eye movements, why was this model not used, or at least compared to the model the authors propose, so that authors could demonstrate the superior nature of their model?

Answer 4: We acknowledge the utility of the Ariadne Toolbox as an open-source software package for studying vestibular stimulation. However, it lacks the specific code necessary for studying MVS, and we noted discrepancies compared to more recent mathematical models. In Arán-Tapia et al., 2023 (<https://doi.org/10.1016/j.compbiomed.2023.107225>), we observed differences in vestibular stimulation during the head impulse test (HIT) compared to previously described literature, including rotational results observed by David et al. (2016). These differences could be attributed to the asymmetric simulation of the cupula

due to the endolymph vortex (Goyens et al., 2019) and the inertial forces directly acting on the elastic cupula during eccentric rotations (Goyens et al., 2020 - <https://doi.org/10.1016/j.heares.2020.108071>). We are uncertain whether the Ariadne Toolbox can capture this complexity in fluid behavior after we carefully evaluate their implemented equations, which are explained in the supplementary material.

This uncertainty led us to choose commercial software like Simcenter STAR CCM+, whose numerical methods and solvers have been validated by numerous research groups and private enterprises, potentially offering more confidence in the results and robustness. We do not claim the superiority of our model designed in this commercial software, as further research and comparison are required. However, based on the good fit of the nystagmus directional components in both our numerical and experimental results, we believe that our MVS model and the one presented in Arán-Tapia et al., 2023 (which share similar solver models for the solid and fluid equations), are capable of representing real vestibular physiological responses observed in humans.

Comment 5: Why were the boundary conditions not manipulated, also to demonstrate robustness of the model? The manuscript in its current form is of limited utility because it is entirely dependent on the single model that the authors present. The model has many parameters, and it is unclear how the behavior would change if the parameters were varied.

Answer 5: Thank you for the suggestion. As you mentioned, we used a set of variables as inputs for the model based on theoretical and experimental observations from the literature. When we designed the model and selected our initial conditions, we conducted a robustness study of the model to qualitatively

evaluate whether the results aligned with expectations from electromagnetics and fluid and solid physical laws. However, we understand that the lack of information in the manuscript may give the impression that the model is limited to these conditions and that we did not perform a proper evaluation.

In light of your recommendation, we have revisited this qualitative analysis in further detail and clarified the robustness of the model quantitatively. Please refer to the new "Model robustness" section in the Supplementary information document. Paragraphs 1 and 2 of the discussion section have been adjusted based on this analysis. The most relevant parameters from this analysis are the subject's age (using hair cell density from the literature as a proxy for age) and the electrical conductivity of different regions in the membranous labyrinth. Hair cell density is expected to decrease proportionally with age, affecting nystagmus components proportionally. This could affect the amplitude of the response but not the pattern of nystagmus, indicating that the sinusoidal behavior observed in Figure 5 is correct regardless of the age considered. Electrical conductivity is only significant when there is a notable difference between endolymph and cupulae conductivities, a scenario not supported by information from the literature.

We hope we have clarified that the model is robust and provides reliable results based on the selected boundary conditions.

Comment 6: Finally, I am unsure of the actual clinical value of the work. The work was done to explain the MVS effect. This effect is not particularly clinically relevant. Individuals may complain about dizziness at the beginning of an MRI, but this typically disappears quickly. What other clinical relevance does this work have?

Answer 6: This study aims to clarify the biophysical mechanisms underlying MVS and to organize the pieces of the puzzle better, as mentioned by Reviewer #2. Specifically, our study sheds light on the expected nystagmus patterns in high-field MRI. It provides biophysical explanations for how nystagmus can be reduced when the patient's head flexes between 27 and 50 degrees, as depicted in Figure 5b. Although these findings are based on our anatomical model, subjects are expected to experience a null point around this head orientation. Prior studies on MVS have discussed how this nystagmus and vertigo can interfere with MRI scans at higher field strengths, contribute to patient safety concerns, and affect the results of functional MRI. A better understanding of this mechanism is expected to lead to optimal solutions to these problems and will be further explored in other studies.

Our model also serves as a foundation for future studies exploring various aspects of vestibular physiology and central adaptation. We view MVS as a novel way of stimulating the vestibular system. Although there are no current clinical applications of this technology, we envision future implications in vestibular rehabilitation and diagnostic testing facilitated by the fundamental understanding provided by this study. We felt these areas were too hypothetical for inclusion in this manuscript.

Comment 7: Figure 1 B and C requires a great deal of imagination or spatial reasoning to envision the orientation of the utricle in relation to the original labyrinth figure. It would be helpful to have a bit more structure of the labyrinth in the background to understand the location of the forces and velocities better.

Answer 7: Thank you for sharing your concern about the complexity of renamed Figures 2b and 2c. Initially, we chose to zoom in on this plane section because it

was necessary to visualize better the magnitude and directions of the Lorentz forces and endolymph velocities. Incorporating the rest of the labyrinth structure directly into these images proved impractical because the transparency of the structures made it challenging to visualize the color map and vectors in both images.

Therefore, we created a new figure called Figure 1c, where this plane can be observed from the same perspective as used in Figures 2b and 2c (left) and from a perpendicular direction to see how this plane represents a slice of the utricle (right). We now consider it easier to envision the location of this plane in the utricle and its relative position to other vestibular structures. We used this plane of section to evaluate the model robustness, as explained in the "Model robustness" section of the Supplementary information document.

Comment 8: How old was the labyrinth? In the supplements I read 55 years but not specifically for the age of the labyrinth. Please make this clear. Are any structural changes expected at that age or from the preservation technique that would affect the results?

Answer 8: We do not have information about the subject's age from whom the micro-CT data was obtained in David et al. (2016). They did not mention this in either the manuscript or the supplementary material. These authors did not mention any differential effect related to the preservation technique based on the specimen's age. Although not formally studied to date, we do not anticipate significant variations in the membranous labyrinth anatomy among young and older adults based on histopathology studies of the membranous labyrinth. Disease processes such as Meniere's disease can affect the anatomy of the membranous labyrinth. Still, the focus in this study has been on the normal

anatomy, which should not meaningfully change with age, aside from hair cell density, as noted now in our supplementary analysis. Therefore, for our model, the age can be considered 55, corresponding to the age selected for the boundary conditions determining the electrical currents. Changing the age for the boundary conditions would be similar to considering another age for the membranous labyrinth anatomy. We have clarified this in the supplementary material section "Boundary conditions."

Comment 9: Consider adding LTS and RTS to the list of abbreviations, instead of simply putting them in the text.

Answer 9: Certainly. We decided to convert these abbreviations to "ear-to-shoulder" (ETS) only because we used it repeatedly. Thank you for pointing that out. We have also ensured all abbreviations were described only once in the text.

Comment 10: The authors say that they did a 3 minute scan of the participants. What data were collected and was it relevant for the study? Were the eye-movements measured during this time, or after, and if they were measured during, would the resulting magnetic fluctuations affect the SPV?

Answer 10: Thank you for your question. First, we should clarify that we did not conduct magnetic resonance imaging of the participants inside the scanner. Instead, we placed them supine on the table of the MRI machine and advanced them into the MRI bore. The effect is due to the static magnetic field, which is continuously on regardless of whether images are being acquired. For this reason, the magnetic field intensity does not fluctuate and is always directed from head to feet in our 7T scanner, as indicated in Figure 1b. We ensured this by using a magnetometer embedded in the video-oculography goggles that

measures the magnetic field continuously, as explained in the methodology section "Infrared video-oculography and head position measurement techniques."

In response to your question about data acquisition, we recorded videos of the eye movements throughout the entire process: before, during, and after entering the MRI bore. Later, we post-processed the videos to measure the three eye movement components (horizontal, vertical, and torsional) and determine the slow-phase velocities (SPV). The most relevant data for the study was the peak nystagmus response observed a few seconds after entering the bore, as illustrated for one of the participants in Figure 4a. After a few seconds in the bore, the nystagmus starts to decline due to neurological adaptation that we have reported previously. Therefore, the analysis of this maximum peak response is the data we used to compare with the numerical results from the crista ampullaris stimulus since it is least affected by central neurological adaptation. We explained this in the methodology sections "Integrated videooculography and head position calibration in MRI" and "Quantification and normalization of nystagmus responses in MRI."

Regarding other data occurring outside of the bore and the time during entering or exiting, we collected the data but did not include it in the analysis as it is irrelevant information to determine the maximum SPV response to magnetic vestibular stimulation. However, all data (before, during, and after) for all participants were consistent with our previous studies, where we observed little or no nystagmus while lying outside the MRI before in darkness, nystagmus increasing to a peak after entering the MRI, then partial adaptation of the nystagmus, followed by a reversal of the nystagmus direction after exiting the

MRI. (Ward et al., 2019 -
<https://journals.physiology.org/doi/full/10.1152/jn.00873.2018> and Pogson et al.,
2023 -
<https://www.frontiersin.org/journals/neurology/articles/10.3389/fneur.2023.1255105/full>). We have added a comment about this in Paragraph 5 of the discussion section to provide more support for our findings. Thank you for bringing up this potential oversight in explanation.

Comment 11: R2 was used in the results, I am assuming to compare the simulations to a sinusoidal curve (since R2 was 1). This was not mentioned anywhere in the manuscript, please make this clear.

Answer 11: We used R-squared to compare the SPV numerical results from simulations to a sinusoidal function. We have clarified this in the last paragraph of the methodology section, "Quantitative modeling of endolymph and cupula dynamics." Thank you for bringing it to our attention.

Comment 12: In the interest of open science, it would be helpful to make your model open source, or combine it with existing resources like Ariadne, please consider.

Answer 12: As mentioned in Answer 4, the model was designed using commercial software, which offers advantages and robustness compared to developing a mathematical model from scratch. However, one of the drawbacks is that our model requires a license to operate and cannot be considered open-source code. Nevertheless, we are open to collaboration and would be willing to share simulation files with interested others upon reasonable request. Additionally, we will consider making this model compatible with Ariadne in the

future or developing a similar software toolbox that includes this model and others with which we are involved. Thank you for your suggestion.

Reviewer #2

Major comment

We appreciate your kind words about our research. Please find below the corrections made to the manuscript based on your recommendations. We have also thoroughly revised the entire manuscript to align the structure with the journal guidelines. In response to the suggestions from Reviewer #1, we have created new Tables 2 and 3 to enhance the comprehensiveness of our research. Additionally, we have included a new Figure 1c and a supplementary material section named "Model robustness" to assess the effect of the boundary conditions selected in our model. We greatly appreciate your detailed review and guidance, and we believe that the text is now more accurate. We hope that we have fulfilled your requirements for publication.

Response to the reviewer's comments

Comment 1: P5 lines 94 to 105 section. Some repetition and needs a bit of re-ordering and re-writing. You also seem to imply that the magnetic field is always head-feet – specifically supine subject in conventional MRI. This is the example but the Lorentz force ideas need explaining more generally. Also say head at rest in the section – so assumed steady state.

Answer 1: We have rewritten Paragraphs 1, 2, and 3 of the results section "Electrical currents and Lorentz force-induced dynamics in the inner ear" to

improve the clarity of our explanation of the Lorentz force effect. The Lorentz force effect is now explained in a general context, followed by a specific description of simulation conditions regarding the B field and electric currents in the utricle. In the last paragraph of this section, we clarified the results observed for the opposite B field direction, feet-to-head case, to indicate that we tested both conditions.

We mentioned "head at rest" in Paragraph 3 to indicate that the endolymph is in a steady state at the beginning of the simulation for this reason. Thank you for pointing that out.

Comment 2: P5 L106 I would prefer the magnetic field to be 'present' rather than 'active'. To me 'active' seems to indicate that we can switch it on and off at will like an evoked response stimulus.

Answer 2: We have replaced "active" with "present" in Paragraph 3 of the results section "Electrical currents and Lorentz force-induced dynamics in the inner ear." This effectively conveys the nature of the static magnetic field in an MRI bore. Thank you for suggesting a more precise term.

Comment 3: P6 L125 (and elsewhere). Probably technically OK to use inhibit/excite but in actuality there is a signal both ways of deflection. I think that these words, to a general reader might confuse – thinking that the information content of the afferent nerve signal stops in some way. Can a very brief reminder be added.

Answer 3: We agree that a general reader might find our description of excitation and inhibition confusing, especially if we use positive and negative signs to

represent the stimulus using the shear strain XY . In reality, this is referenced to the resting state of the hair cell.

We have added a comment in Paragraph 2 of the methodology section, "Quantitative modeling of endolymph and cupula dynamics" to clarify that excitation and inhibition are referenced to the resting state signal of the afferent nerve fibers.

Comment 4: P21 L386 '... given the physical geometrical constraints of the human body and MRI machines.' Or something similar.

Answer 4: Thank you, this is much clearer. We have incorporated your suggestion into Paragraph 9 of the discussion section.

Comment 5: P28 L563 Add somewhere that this electrodynamic problem is firmly in the quasi-static regime and can be solved with non-full-wave EM finite-element or boundary-element methods. Also, you didn't actually say how you did this simulation from a computational point of view.

Answer 5: We have now included additional information on how we designed the simulation, explicitly stating that we operate within a quasi-static regime and utilize non-full-wave EM methods. Please refer to Paragraphs 4, 5, and 6 of the methodology section "3D computational modeling of the membranous labyrinth" and Paragraphs 2 and 3 of the methodology section "Quantitative modeling of endolymph and cupula dynamics" for details.

We hope that this clarification makes the content more useful to a broader group of readers. Additionally, please note that in the methodology, we refer to our previous research (Arán-Tapia et al., 2023 - <https://doi.org/10.1016/j.compbimed.2023.107225>), where the fluid-solid

problem modeling is more detailed. Furthermore, the supplementary information provides extra information for readers interested in the electromagnetic boundary conditions and robustness of the model.

Comment 6: P31 L637 The canals and cupulae are not-orthogonal, so your statement here can't be true. But presumably you are arguing that within natural variation of subject response, this can be assumed.

Answer 6: Thanks for the comment. We agree that the statement will not be exactly true for most individuals, but we decided to build a representative model with those assumptions. We have updated Paragraph 15 of the methodology section titled "Quantitative modeling of endolymph and cupula dynamics" accordingly.

Comment 7: P33 L691 'scan' is not what you are doing (I hope with all that equipment in there!). Use 'exposure' or 'trial' as elsewhere.

Answer 7: Of course, that would not be a good idea! Thank you for bringing it to our attention. To view how we rewrote the terms, please refer to Paragraph 2 of the methodology section "Integrated videooculography and head position calibration in MRI".

Comment 8: P35 L738 add '...as expected in accordance with manufacturer's standard installation practice'. It isn't a random thing or mistake.

Answer 8: Certainly, we have incorporated your suggestion. Thank you. Please refer to Paragraph 3 of the methodology section " Quantification and normalization of nystagmus responses in MRI".

Comment 9: P34. You quite rightly discuss baseline correction of data. Any comment on the fact that the subject has already adapted to the magnetic field present (0.7 T for 7T magnet). The additional field exposure would be measured at peak response, but the 0.7T is measured at the lower steady state. Would be about 5% error perhaps. Just that you are subtracting out 50 microtesla effects – which are presumably just resting-state subject effects.

Answer 9: Yes, addressing this aspect is indeed challenging. We acknowledge that adaptation during the initial minutes spent outside the MRI, particularly outside the 7T, could also be significant when measuring the peak response in MVS-induced nystagmus. Additionally, we observed significant nystagmus in some subjects during the resting state (under the Earth's magnetic field effect) that we considered necessary to take into account.

There are two main reasons why we decided to measure the peaks with respect to the baseline (Earth's magnetic field) rather than the measurements taken out of the bore. Firstly, the complex effect of the "curved" magnetic field outside the bore could potentially affect the labyrinths differently compared to the head-to-feet or feet-to-head orientations inside the bore. Secondly, this effect would vary between each MRI machine's specifications, making it less efficient when comparing normalized values, as they would not be relative to the same baseline effect.